

# Impact of typhoons on the composition of the upper troposphere within the Asian summer monsoon anticyclone: the SWOP campaign in Lhasa 2013

Dan Li[1,2], Bärbel Vogel[2], Jianchun Bian[1,3], Rolf Müller[2], Laura L. Pan[4], Gebhard Günther[2], Zhixuan Bai[1,3], Qian Li[1], Jinqiang Zhang[1], Qiujun Fan[1,3], and Holger Vömel[5]

[1]Key Laboratory of Middle Atmosphere and Global Environment Observation (LAGEO), Institute of Atmospheric Physics, Chinese Academy of Sciences, Beijing, China
[2]Institute of Energy and Climate Research: Stratosphere (IEK-7), Forschungszentrum Jülich, Jülich, Germany
[3]College of Earth Science, University of Chinese Academy of Sciences, Beijing, China
[4]Atmospheric Chemistry Observations & Modeling, National Center for Atmospheric Research, Boulder, CO, USA
[5]Earth Observing Laboratory, National Center for Atmospheric Research, Boulder, CO, USA

*Correspondence to:* Dan Li (lidan@mail.iap.ac.cn)

**Abstract.** In the frame of the SWOP (sounding water vapour, ozone, and particle) campaign during the Asian summer monsoon (ASM), ozone and water vapour profiles were measured by balloon-borne sensors launched from Lhasa (29.66° N, 91.14° E, elevation 3,650 m), China, in August 2013. In total, 24 soundings were launched, nearly half of which show some strong variations in the relationship between ozone and water vapour in the tracer−tracer correlation in the upper troposphere and

5    lower stratosphere (UTLS). 20-day backward trajectories of each sounding were calculated using the trajectory module of the Chemical Lagrangian Model of the Stratosphere (CLaMS) to analyse these variations. The trajectory calculations demonstrate that three tropical cyclones (tropical storm Jebi, typhoons Utor and Trami), which occurred over the Western Pacific Ocean during August 2013, had a considerable impact on the vertical distribution of ozone and water vapour by uplifting marine air masses to altitudes of the ASM anticyclone. Air parcels subsequently arrived at the observation site via two primary pathways:

10   firstly via direct horizontal transport from the location of the typhoon to the station within approximately three days, and secondly via rotational subsidence, during which air parcels descend slowly along a circle following the anticyclone flow within a timescale of one week. Furthermore, the interplay between the spatial position of the ASM anticyclone and tropical cyclones plays a key role in controlling the transport pathways of air parcels from the boundary layer of the Western Pacific to Lhasa in horizontal as well as vertical transport. Moreover, the statistical analysis shows that the strongest impact by typhoons

15   is found at altitudes between 14.5 km and 17 km (365−375 K). Low ozone values (50−80 ppbv) were observed between 370 K and 380 K due to the strong vertical transport within tropical cyclones.

## 1    Introduction

The Asian summer monsoon (ASM) anticyclone is one of the largest upper-level circulation systems that spans from Southeast Asia to the Middle East in the upper troposphere-lower stratosphere (UTLS) during the boreal summer (Mason and Anderson,



1963). The ASM is considered to be coupled with persistent monsoonal convection over the South Asia region during the summer season (Randel and Park, 2006; Randel et al., 2016). The longitudinal distribution of the core of the ASM anticyclone has two preferred modes, referred to as the Iranian Mode and the Tibetan Mode, respectively (Zhang et al., 2002). This bimodality has an impact on the distribution of atmospheric trace species (such as water vapour and ozone) and dynamic parameters in the UTLS region (Yan et al., 2011). The total column ozone (TCO) over the Tibetan Plateau in the boreal summer is lower than that of other regions at the same latitude (Zhou and Luo, 1994). This phenomenon is termed as the "summertime ozone valley". The transport associated with the ASM circulation is one of the most critical causes of the summertime ozone valley over the Tibetan Plateau (Tian et al., 2008; Bian et al., 2011).

The ASM is recognised as an important transport pathway for boundary layer air enriched in greenhouse gases or pollutants (e.g., water vapour, hydrogen cyanide (HCN) produced by biomass burning, carbon monoxide, aerosol etc.) to enter the stratosphere (e.g., Park et al., 2009; Randel et al., 2010; Chen et al., 2012; Vogel et al., 2015). According to trajectory simulations, air within the ASM anticyclone is impacted by surface sources from various regions: the Western Pacific Ocean, India and Southeast Asia, Eastern China, the Tibetan Plateau, and the Indian Ocean (Chen et al., 2012; Bergman et al., 2013; Vogel et al., 2015; Tissier and Legras, 2016). Using trajectory calculations during the summers of 2001−2009, Chen et al. (2012) showed that 38 % of the air mass at tropopause height within the ASM region is from the Western Pacific region and South China Sea. Bergman et al. (2013) demonstrated that 10 % of the air mass in the anticyclone at 100 hPa originated from the Western Pacific in August 2001. Using artificial emission tracers in CLaMS (the Chemical Lagrangian Model of the Stratosphere) in summer 2012, Vogel et al. (2015) found a strong variability in contributions from Southeast Asia to the composition of the ASM anticyclone during the monsoon season at 380 K. The major transport processes for tropospheric tracers transporting from boundary layer sources to the upper troposphere and lower stratosphere include both deep convection with rapid vertical transport and large-scale slow upward circulation within the ASM anticyclone (Yan and Bian, 2015; Ploeger et al., 2015b). An additional transport pathway from the boundary layer to the edge region of the ASM is strong uplift in typhoons and subsequently entrainment into the circulation of the anticyclone (Vogel et al., 2014).

Strong tropical cyclones in the western part of the Pacific Ocean are also known as typhoons. Tropical cyclones vary in the horizontal scale over a wide range from 100 km to 2,000 km, characterized by spiral rain-bands which consist of bands of cumulus convection clouds (Emanuel, 2003). These bands of clouds are often accompanied and sometimes dominated by strong updrafts, whereas downdrafts occur concomitantly between these convective clouds. The vertical typhoon circulation with strong uplift can lift marine boundary layer air masses into the UTLS region (Vogel et al., 2014; Minschwaner et al., 2015). The tropical Pacific and Western Atlantic are situated far away from anthropogenic emissions and biomass burning sources, and are thus characterized by low ozone values in the lower troposphere (Thompson et al., 2003; Jenkins et al., 2013). Therefore, low ozone concentrations are sometimes measured in the upper troposphere within typhoons or hurricanes (Cairo et al., 2008). Using balloon-borne ozone data over Socorro (North America), Minschwaner et al. (2015) show how a hurricane uplifts the eastern/central tropical Pacific boundary air with extremely low ozone to the upper troposphere. Meanwhile, the downdraft in typhoons can transport ozone-rich air from the lower stratosphere down to the troposphere (Das et al., 2016b) and



even to the surface (Jiang et al., 2015). Thus, tropical cyclones exert critical impact on air masses and energy transport between the surface and the UTLS region (Fadnavis et al., 2011; Venkat Ratnam et al., 2016).

In several model studies, contributions of air masses originating from the Western Pacific are found within the ASM anticyclone region (e.g., Park et al., 2009; Chen et al., 2012; Bergman et al., 2013; Vogel et al., 2015; Tissier and Legras, 2016) during summer. Furthermore, in the Western Pacific belt, tropical cyclones reach peak activity in late summer (Emanuel, 2003). Several previous studies have suggested an impact of deep convection by tropical cyclones, in particular typhoons, on the chemical composition of the ASM anticyclone (e.g., Li et al., 2005; Munchak et al., 2010; Bergman et al., 2013; Vogel et al., 2014). In spite of the many satellite measurements and model simulations, the transport process of air masses from the planetary boundary layer of the Western Pacific Ocean to the ASM anticyclone associated with typhoons is still unclear and requires further investigations. In-situ measurements of the atmospheric chemical compositions over the Tibetan Plateau are sparse (Zheng et al., 2004; Bian et al., 2012). In this study, balloon measurements with high vertical resolution from Lhasa in August 2013 provide highly accurate water vapour and ozone profiles from the surface to the lower stratosphere. Combining these in situ measurements with trajectory calculations performed using the CLaMS model (McKenna et al., 2002; Pommrich et al., 2014, and references therein) is an ideal method for analysing the source regions and transport pathways of air masses affected by tropical cyclones.

The goal of this investigation is both to identify transport pathways from air masses uplifted by tropical cyclones into the ASM anticyclone and to quantify their impact on ozone observed in the upper troposphere over the Tibetan Plateau in August 2013. This paper is organized as follows: Sect. 2 describes the balloon sonde data and the trajectory calculation with CLaMS. In Sect. 3, we present the tropical cyclones that might have affected the composition in Lhasa during summer 2013. In Sect. 4, we focus on model results as well as the spatial position interplay between ASM and cyclones. will be displayed . In the final section, we summarize the results and present our conclusions.

## 2  Data and Model description

### 2.1  Data

To investigate the spatial variability of the UTLS ozone concentration and water vapour in the ASM anticyclone, the SWOP (sounding water vapour, ozone, and particle) campaign was conducted by the Institute of Atmospheric Physics, Chinese Academy of Sciences, during the summer monsoon period. Balloon sondes were launched in Lhasa (29.66° N, 91.14° E, above sea level (asl.) 3,650 m), China, in August 2013. Lhasa is located on the Tibetan Plateau, one of the source regions of air masses found within the ASM anticyclone. More detailed information of the Lhasa site is provided by Bian et al. (2012, in Fig. 1). A total of 24 soundings were launched at a rate of once per day around 22:30 BST (Beijing Standard Time, UTC+8) from 4 to 27 August 2013. The serial number of balloon sondes, dates, times, and burst altitude for each balloon are listed in Table 1. All but for three ascents reached altitudes of 26.5 km or more.

The balloon-borne payloads consist of a cryogenic frost point hygrometer (CFH) (Vömel et al., 2007b), an electrochemical concentration cell (ECC) ozonesonde, and an iMet radiosonde to measure profiles of water vapour, ozone, and routine mete-



orological variables (pressure, temperature, relative humidity, and winds), respectively. On average, a balloon ascent lasts for $\sim 75$ minutes from the surface to about 30 km altitude before bursting. The measurement uncertainty of CFH is less than 9 % in the tropopause region (Vömel et al., 2007a), and the ozone uncertainty estimated by Smit et al. (2007) is better than $5-10$ %. The ECC sensor response time is about 22 s in the troposphere (Selkirk et al., 2010) and the balloon ascent rate is $\sim 4-6\,\mathrm{m\,s^{-1}}$.

As a result, the ozone and water vapour mixing ratios are provided with a 100 m vertical resolution from the surface through 50 hPa, to attenuate the instrument response time effect (Minschwaner et al., 2015).

A scatter plot of the water vapour and ozone of all balloon flights in August 2013 is shown in Fig. 1. A double-logarithmic scale is used to highlight the correlation between water vapour and ozone in the UTLS region. The two tracers typically have an L-shaped correlation in tracer−tracer space (Pan et al., 2007; Bian et al., 2012; Pan et al., 2014), but some profiles reveal

relatively low ozone or low water vapour in the UTLS region (see profiles that are marked in colour in Fig. 1) compared to all other profiles measured in Lhasa during August 2013.

The observation on 11 August 2013 (Fig. 1, in red) shows a distribution, with low ozone and low water vapour in the corner of the "L" (corresponding to the UTLS region), but with very high ozone in the troposphere, where water vapour has a concentration between 100 ppmv and 500 ppmv. The transport processes leading to these conditions will be studied in section

4.1.2. On the next day, a profile (marked in green) demonstrates particularly low water vapour in the troposphere. Another three profiles also show low water vapour and low ozone structure in the UTLS region during the period of $23-25$ August. High ozone ($> 100$ ppbv) was observed when water vapour mixing ratios were greater than 30 ppmv on 19 August. The extremely low ozone ($\sim 50$ ppbv) and low water vapour ($\sim 5$ ppmv) are displayed in the corner of the "L" from 23 to 25 August 2013. A similar correlation has also been reported by Bian et al. (2012) based on measurements in Kunming ($25.01°$ N, $102.65°$ E, asl.

1,889 m), China, in 2009. Munchak et al. (2010) show that low ozone near the tropopause is caused by rapid ascent in oceanic deep convective systems associated with a tropical typhoon. In the same way, several profiles measured in Lhasa in 2013 are impacted by different tropical cyclones. The regions of three profiles impacted by cyclones, which were measured on 11, 19, and 24 August 2013, are highlighted in Fig. 1 (inset) and will be discussed in more detail in section 4.1.

## 2.2   Model

In order to investigate the distributions with low ozone and low water vapour in the UTLS region, the trajectory module of the CLaMS model (McKenna et al., 2002; Pommrich et al., 2014, and references therein) was used to calculate 20-day backward trajectories along each balloon's ascent flight path in Lhasa in 2013. The CLaMS model is particularly well-suited for the simulation of tracer transport in the vicinity of strong transport barriers and the associated tracer gradients such as the polar vortex (e.g., Müller et al., 2005; Günther et al., 2008), the extratropical tropopause (e.g., Vogel et al., 2011; Konopka and

Pan, 2012), and the Asian monsoon anticyclone (e.g., Ploeger et al., 2015a; Vogel et al., 2015). CLaMS was applied to analyse aircraft and balloon measurements with a focus on stratospheric chemistry (e.g., ozone loss processes) (e.g., Grooß and Müller, 2007) and the transport of trace gases (e.g., water vapour and ozone) in the UTLS (e.g., Vogel et al., 2016).

The model was driven by dynamic fields from the European Centre for Medium-range Weather Forecasts (ECMWF) re-analysis interim (Era-Interim) (Dee et al., 2011). The input data are recorded at 6-hour intervals on a regular grid with $1° \times$





1° in latitude/longitude on hybrid levels (60 levels from 1013.25 hPa to 0.1 hPa). The trajectories were calculated with vertical velocities from the diabatic heating rate when the pressure is less than 300 hPa, and using a pressure-based hybrid vertical coordinate $\zeta$ when the pressure is higher than 300 hPa (Pommrich et al., 2014; Ploeger et al., 2010).

To further investigate the boundary sources of air parcels near the tropopause layer in Lhasa, parcels were selected according

to the following criteria: parcels that reach the lower troposphere (LT) within 20-day backward trajectories are referred to as "target air parcels". Here, the top of LT is defined as $\zeta < 190$ K (approximately 3.0 km above the surface). $\zeta$ as a terrain-following coordinate is universally applicable for keeping the vertical spacing between the Earth's surface and the top of LT constant. On the basis of this criterion, parcels reaching the upper troposphere and lower stratosphere within 20-day backward trajectories are eliminated. As a result, we focus solely on the air parcels that reach to the LT and are uplifted to the location of

the measurement.

## 3   Tropical cyclones in August 2013

After analysing the 20-day backward trajectories from the output of CLaMS, we found that three tropical cyclones (named Trami, Jebi, and Utor), which occurred over the Western Pacific, influence the composition in middle/upper troposphere at the Lhasa site. Table 2 shows the name of the cyclones, their duration, intensity, peak, and the Saffir−Simpson hurricane wind scale

(SSHWS) index. Typhoon Utor is the strongest of the three typhoons with an intensity of 925 hPa, a peak speed of 195 km/h, and an SSHWS index of 4, the highest of the three tropical cyclones (for more details see [1]).

Typhoon Trami developed east of Taiwan on 16 August 2013. During the next two days it moved towards the southeast, continued to gain in strength, and was upgraded to a tropical storm. Trami turned northward on 19 August and then turned northwestward on 20 August. After moving west-northwestward over the East China Sea, Trami hit Fujian province, China,

on 21 August. Over the next couple of days, it continued to pass through the Jiangxi and Hunan provinces of China. Finally, Trami dissipated over Guangxi on 24 August (see green dots in Fig. 2a, the blue points will be discussed in section 4.3).

On 28 July 2013, tropical depression Jebi formed near the southern coast of Luzon Island. As it moved northwestward, it was observed crossing the Philippines and reached the South China Sea on 30 July before it continued to move northwestward. After crossing the northern part of Hainan Island and the Gulf of Tonkin, Jebi made landfall over Northern Vietnam on 3

August, and dissipated several hours later (track and date are marked in Fig. 2b).

Typhoon Utor formed northwest of the Yap Islands on 8 August 2013. As this system moved westward, it developed rapidly and was upgraded to typhoon intensity on 10 August. Turning west-northwestward on 11 August, Utor reached its peak intensity (SSHWS, category 4). After hitting Luzon Island, it maintained its typhoon intensity over the South China Sea. The system then tracked northward and Utor made landfall over Yangjiang in Guangdong, China, on 14 August. On next day, Utor

weakened into a tropical depression. However, the remnants began tracking very slowly in the Guangxi region until the tropical depression finally dissipated on 18 August (see Fig. 2c).

---

[1]websites: http://agora.ex.nii.ac.jp/digital-typhoon/year/wnp/2013.html.en and http://www.nrlmry.navy.mil/tcdat/tc13/WPAC/





According to 20-day backward trajectory calculations, thirteen profiles were impacted by these three Western Pacific tropical cyclones (marked in the last column of table 1). From 11 to 13 August, tropical storm Jebi transported air parcels to the Lhasa site. Typhoon Utor had a long-term impact on the profiles during the period of 17−26 August. The last four profiles were impacted both by typhoons Utor and Trami. Three profiles highlighted in bold in Table 1 (11, 19, and 24 August) were
influenced by different tropical cyclones and will be discussed in detail in section 4.1.

## 4  Results

To obtain further insight into the impact of tropical cyclones on ozone and water vapour in the upper troposphere in Lhasa, we analyse three ozone and water vapour profiles (bold in table 1) as a case study. The results shown below are based on ozone and water vapour profiles observed at the Lhasa site on 11, 19, and 24 August associated with 20-day backward trajectories
from the CLaMS model. In addition, the meteorological conditions that caused the transport of air parcels from the Western Pacific to Lhasa are analysed.

### 4.1  Analyses of three selected cases

#### 4.1.1  Case 1 (Trami)

Figures 3a−c show the CLaMS backward trajectories and measured profiles of ozone and water vapour influenced by typhoon
Trami. The mean ozone profile is obtained by averaging individual profiles over Lhasa in August 2013 (grey line in Fig. 3b). At 14:13 UTC on 24 August, very low ozone mixing ratios of about 50 ppbv (lower than the average of ozone mixing ratios) and low water vapour mixing ratios of about 7 ppmv (lower than other water vapour values in Fig. 1) were measured just below the tropopause (the World Meteorological Organization (WMO) tropopause altitude is 17.6 km) in a 14−16.5 km layer.

20-day backward trajectories initialized in Lhasa on 24 August (Figs. 3a and c) show that air parcels with low ozone con-
centration originated from the boundary layer of the Western Pacific Ocean. Most of the air parcels were lifted up to $\sim 17$ km through the strong upward airflow associated with typhoon Trami. When these parcels arrived at the tropopause region, they encountered the cold upper troposphere in the Western Pacific Ocean. The minimum temperature of each parcel ranged from $-82.3°$ C to $-72.7°$ C (Fig. 3a). Using the equation (7) of Murphy and Koop (2005), we calculate the vapour pressure over ice. The water vapour mixing ratios are obtained by using vapour pressure divided by air pressure, with a range of 3.1 ppmv to
12.4 ppmv for the above-mentioned temperature range. This value range is in good agreement with the water vapour measured in this layer, indicating that the air parcels were dehydrated when they passed through the cold upper troposphere of the Western Pacific. Air parcels originated from the marine boundary layer, moving rapidly upward where the process of strong uplift occurred during 18 to 21 August (Fig. 3a). Furthermore, the value of potential vorticity (PV) is lower than 2 PVU during the period of strong upward transport, indicating rapid uplift of boundary layer air masses to the upper troposphere (Fig. 4a).
Air parcels with low ozone travelled 3,000 km horizontally within three days from the top of typhoon Trami to the Lhasa site, with PV increasing slowly. This indicates the rapid and direct influence of typhoon-induced transport on ozone mixing ratios





and water vapour in the upper troposphere over Lhasa within a short timescale. Low ozone mixing ratios above 200 hPa were also observed right above the typhoon's track over the Western Pacific using Aura's Ozone Monitoring Instrument data (Fu et al., 2013). The reason for this is the strong upward propagation of air masses associated with deep convection in typhoons. Minschwaner et al. (2015) also demonstrate that meteorological conditions associated with tropical cyclones have a strong

influence on ozone mixing ratios in the upper troposphere over timescales of 3−5 days.

### 4.1.2   Case 2 (Jebi)

20-day backward trajectories of target parcels influenced by tropical storm Jebi on 11 August 2013 are shown in Fig. 3d. Most air parcels are uplifted from the LT and the planetary boundary layer to altitudes just below the tropopause within a time scale of more than one week. However, backward trajectories of three air parcels show a very rapid uplift during the period of

1−3 August. These air parcels reach a maximum altitude of approximately 16 km. The parcels also pass through a region of very low temperatures ($-75°$ C) in the upper troposphere above the tropical Western Pacific where they dehydrate. These air parcels are very dry with water vapour mixing ratios of $\sim$8 ppmv. The PV values associated with these air masses are lower than 2 PVU (1 PVU=$10^6$ K m$^2$ kg$^{-1}$ s$^{-1}$) during ascent. After the parcels arrived at the maximum altitude, the parcels' PV increased, especially for the parcels located at a higher altitude, with PV values greater than 2 PVU (Fig. 4b).

On 11 August 2013, the parcels reached an altitude of 12−15 km (which is much lower than the WMO tropopause altitude of 17.25 km) over Lhasa with a slow downwelling within approximately one week. Low ozone and low water vapour are observed in this layer (Fig. 3e). These measurements indicate that tropical storm Jebi also lofted air with low ozone from the LT of the tropical Pacific to Lhasa. A very high ozone peak was also observed in the 10.5−12 km layer, with an ozone enhancement at 100 ppbv. The reason for this observation is that the air originating from the lower stratosphere with high ozone concentration

intruded into this layer (not shown here).

The three-dimensional transport pathways of air parcels are displayed in Fig. 3f. Air parcels are lifted from the Western Pacific to near the tropopause layer through strong vertical air flow associated with the tropical storm Jebi. The parcels are then transported around the ASM anticyclone within a time scale of one week. Vogel et al. (2014) showed in a case study that boundary layer air masses originating in Southeast Asia were rapidly uplifted within typhoon Bolaven. For the case study

shown here, the vertical transport mechanism of typhoon Jebi are similar to that found in typhoon Bolaven (Vogel et al., 2014). All ozone and water vapour profiles observed on 12 and 13 August 2013 are impacted by tropical storm Jebi (table 1). The backward trajectories are similar to those for 11 August (trajectories not shown here).

### 4.1.3   Case 3 (Utor)

In contrast to case 1 and case 2, the vertical profiles of ozone and water vapour show a remarkable laminar structure in the

middle and upper troposphere on 19 August (Fig. 3h). Ozone mixing ratios display a high value (low value) between 9.3 km and 10.2 km (10.2−11.2 km). In contrast, the water vapour mixing ratio here exhibits local low value (high value). Above 11 km, the ozone values are much larger than the average of total ozone profiles except in a layer of 13.2−14.5 km. Many factors such



as convection (Thompson et al., 2010), gravity or Rossby wave (Thompson et al., 2011), and stratospheric intrusions (e.g., Das et al., 2016a) can cause the laminated vertical structure of ozone in the middle/upper troposphere and lower stratosphere.

The CLaMS trajectory calculations show that two clusters of air parcels originating from the surface can be traced under the influence of typhoon Utor (Figs. 3g and i). Target parcels at higher altitude in the upper troposphere originate from the Western Pacific Ocean and are first uplifted to an altitude of 14 km within two days. The maximum altitudes of target parcels under the uplift effect of typhoon Utor are lower compared to that of cases Trami and Jebi. Secondly, air parcels reach the maximum altitude at 18:00 UTC on 12 August after slow upwelling. Subsequently, these target air parcels were transported quasi-horizontally and arrived at the Lhasa observatory on 19 August. When the air parcels reached the highest altitude, the PV increased strongly within about three days (Fig. 4c). Explaining air parcels maintain high ozone concentration at an altitude of 15 km (Fig. 3h).

Target parcels at lower altitude are transported from the lower troposphere to the middle troposphere by convective uplift associated with the landfall of Utor (Figs. 3g and i), causing the low ozone and high water vapour observed in the middle troposphere. Our findings show that typhoon Utor pumped air parcels from the boundary layer to the upper and middle troposphere both before and after landfall.

## 4.2 Interplay between ASM and tropical cyclones

To better understand the interplay between the ASM anticyclone and typhoons, the meteorological conditions are analysed in detail for the cases considered here. Figures 5−7 show the geopotential height of the ASM anticyclone at 100/150 hPa pressure level and the sea-level air pressure of the three tropical cyclones. The ASM anticyclone is characterized by a pronounced east−west oscillation of the location of the core of the ASM anticyclone: the Tibetan mode and the Iranian mode (see Fig. 6). The bimodality in the location of the anticyclone has been found in previous earlier studies at 100 hPa for daily data (Yan et al., 2011), for pentad (5-day) mean data (Zhang et al., 2002), and for monthly mean data (Zhou et al., 2009; Nützel et al., 2016).

At 06:00 UTC on 20 August 2013, typhoon Trami is located right below the southeastern edge of ASM anticyclone circulation, with most of the target parcels located inside the typhoon characterized by low potential temperature (∼ 330 K) (Fig. 5a). Thirty-six hours later, the parcels have reached at altitudes of the ASM (∼ 360 K) due to the strong upward propagation of deep convection in typhoon Trami (Fig. 5b). This strong uplift can be seen clearly in Fig. 3a. After the parcels entered the ASM, they traveled along the easterly wind flow on the south side of the ASM anticyclone. The interplay between the spatial position of the ASM anticyclone's circulation and the typhoon creates conditions that can rapidly pump air parcels with low ozone from the marine boundary layer to an altitude of 16 km. The parcels are then dehydrated and transported to Lhasa in a short timescale. It is for the precise reason that the entire process (uplifting and horizontal transport) occurs on a short-time scale of about one week that the low ozone mixing ratios and low water vapour can be detected at the Lhasa site.

Focusing on the Jebi case, the ASM circulation differs from the Trami case. It has two centres, with the main core located over the Iranian Plateau and the weak center located over the Western Pacific. The Lhasa site is located on the eastern side of the primary center of the ASM anticyclone (Fig. 6). Parcels from the marine boundary layer are lifted by convection associated with tropical storm Jebi and are then entrained into the southeast edge of the ASM anticyclone (Figs. 6a and b). During the





uplift process, the potential temperature of parcels increases slowly. The westward flow of the ASM moves the parcels from right above the cyclone's track ($\sim 110°$ E) to the western edge of the ASM ($\sim 20°$ E). They subsequently move clockwise around the ASM, before they arrive at the Lhasa site (Figs. 6c−e). The potential temperature of the air parcels increases to 370 K. Vogel et al. (2014) show a similar transport pathway around the outer edge of the ASM anticyclone.

At 18:00 UTC on 12 August, the core of the ASM was located over the Iranian Plateau; the position of typhoon Utor was disconnected from the position of the ASM. Three days later, the core of the ASM was divided into two centres with the primary core located over the Tibetan Plateau. Utor made landfall next to the southeast edge of the main core of the ASM. At the same time, high latitude air with high PV intruded equatorward between the southeast edge of the ASM and the northwest edge at the top of typhoon Utor, forming a thin filament at the east side of the Tibetan Plateau. The strong horizontal wind

shear caused the mixing processes that occurred between air masses from the UTLS region at high latitude with high PV and high ozone concentrations and air masses from the lower troposphere of the North Pacific with low ozone mixing ratios. As a result, we observed high ozone concentrations in laminar structures in the upper troposphere at the Lhasa site (see Fig. 3h). A detailed analysis of mixing processes affecting ozone and water vapor is beyond the scope of this study. However, in-mixing of air mass from the stratosphere influences the structure of ozone profiles observed at the Lhasa site. From 12:00 UTC on 18

August, the parcels are transported around the low pressure system, which is located at around 110° E, 28° N (see Fig. 7c), before slowly moving to Lhasa. The spatial interplay between the position of the primary core of the anticyclone and typhoons is a key factor influencing the transport pathways of air parcels from the top of typhoons to the Lhasa site.

The total attenuated backscatter profiles at 532 nm from CALIOP (Cloud-Aerosol Lidar with Orthogonal Polarization) lidar (Rogers et al., 2011) along the CALIPSO (Cloud-Aerosol Lidar and Infrared Pathfinder Satellite Observation) orbit tracks

(purple line in Figs. 5b, 6c, and 7a) are shown in Fig. 8, with the vertical velocity from Era-Interim and the altitude of the target parcels. The figure shows that the top of deep convective clouds associated with tropical cyclones is $\sim 17$ km at 18:00 UTC on 21 August 2013 (near typhoon Trami), $\sim 16.5$ km at $\sim 06:00$ UTC on 3 August 2013 (near tropical storm Jebi), $\sim 16$ km at 18:00 UTC on 12 August 2013 (near typhoon Utor). The deep convection associated with the three tropical cyclones extends into the ASM anticyclone. The altitude of the cloud top is close to the highest altitudes to which the air parcels are uplifted by

typhoons Jebi and Trami (Figs. 8a and c).

### 4.3   Impact of Tropical cyclone on UT ozone

The average geographical locations for which strong uplift ($> 9$ K/day) of target air parcels occurred under the influence of tropical cyclones are shown in Fig. 2. The target air parcels under the influence of typhoon Trami analysed in our case study experienced strong upward transport over the Pacific Ocean. Rare air target parcels were detected after Trami made landfall.

The reason for this was because no further balloon measurements were available after the end of the campaign on 27 August 2013 (Fig. 2a). The geographical location of the strong uplift of parcels moves along the track of tropical storm Jebi over the region from the Western Pacific to South China (Fig. 2b). Furthermore, typhoon Utor exerts a major influence on the uplifting of most target parcels after its landfall. This is caused by the long lifetime of Utor after landfall. In contrast, only a few target parcels are detected in Lhasa that experienced strong uplift before Utor's landfall (Fig. 2c).





Figure 9 shows the relative frequency distribution of target air parcels originating from the LT; each colour bar represents the results with or without the influence of tropical cyclones. The relative frequency is calculated as the fraction, which is defined as the number of target air parcels within each layer divided by the total number of all parcels of each layer sampled in Lhasa (altitude range: $7-19$ km/$330-390$ K, thickness of each layer is 500 m/2.5 K). The probability distribution of the target parcels

influenced by typhoons (red colour bars) shows two peaks: the first peak at around 15.5 km/365 K and a second peak at around 10 km/350 K. In contrast, the target parcels uplifted without the impact of typhoons (e.g. by convection; blue colour bars) show a broader distribution at altitudes of $\sim 7-13$ km/$340-360$ K. It is clear that only the target parcels impacted considerably by typhoons show a clear peak structure in the layer of $14.5-17$ km/$365-375$ K. This finding indicates that strong vertical transport within typhoons over the ocean can pump air parcels from the boundary layer to a higher altitude than convection

over land.

In figure 10, the frequency distributions of all ozone profiles observed in August 2013 are shown at four isentropic layers in order to demonstrate that ozone values near the tropopause could be influenced by typhoons. The relative frequency is calculated as the fraction of the number of air parcels with each bin of ozone value (10 ppbv in the layers of $360-370$ K and $370-380$ K, 5 ppbv in the layers of $340-350$ K and $350-360$ K) divided by the total parcels of each layer. Ozone with or

without the influence of tropical cyclones is identified as the parcels from Fig. 9b. The highest layer, between 370 K and 380 K potential temperature, shows three ozone maximum with the major ozone maximum of 0.2 between 120 ppbv and 140 ppbv (in black). The maximum with low ozone value ($50-80$ ppbv, in red) is associated with strong vertical transport with the significant effects of tropical cyclones over potential temperature levels $\sim 370-380$ K (Fig. 10a). Distributions of ozone in Lhasa shift toward lower concentrations in the layers of $360-370$ K and $340-350$ K (Figs. 10b and d). However, the ozone

concentration in the layer of $350-360$ K show a wide distribution with ozone of $65-110$ ppbv. This layer is influenced by convective outflow of the ASM region (Qie et al., 2014; Yan and Bian, 2015) and is also a region penetrated by stratospheric intrusions (Fadnavis et al., 2010). As a result, air masses with both low and high ozone concentration are transported to the layer of $350-360$ K, resulting in the strong variability observed.

## 5   Summary and conclusions

High-resolution ozone and water vapour profiles over Lhasa, China were measured once per day over the period from 4 to 27 August 2013 in the frame of the SWOP campaign. Half of them show a strong variability in the correlation between ozone and water vapour mixing ratios in the upper troposphere region. These relationships were investigated using CLaMS trajectory calculations driven by ERA-Interim reanalysis data. We find that tropical storm Jebi and typhoons Utor and Trami, which occurred over the Western Pacific during this period, had a strong impact on the vertical structure of ozone and water vapour

profiles measured in Lhasa. Tropical cyclones pump air masses from the lower troposphere and the planetary boundary layer up to altitudes close to the tropopause region. Air parcels uplifted by tropical cyclones can reach Lhasa via two different horizontal long-range transport pathways: (a) Direct horizontal transport: the parcels travelled from the top of the typhoon to the Lhasa



site directly within about three days, and (b) Transport around the ASM anticyclone: the parcels are transported around the ASM anticyclone within a timescale of more than one week.

Transport pathways of air parcels depend strongly on the relative spatial position of the main core of the ASM anticyclone and the tropical cyclones. A relatively close position between the ASM and typhoon Trami led to the parcels arriving at the Lhasa site in a short period of time, passing through the very cold upper troposphere over the Western Pacific Ocean. As a result, low ozone and low water vapour structures can be observed clearly at the Lhasa site. The timing as well as the spatial colocation between the ASM anticyclone and the tropical cyclone determine whether target air parcels from the boundary layer can reach the upper troposphere over Lhasa.

Tropical cyclones have a different impact on the vertical transport of air parcels from the boundary layer up to the upper troposphere. We found that typhoon Trami had an important effect on the strong uplift of air parcels measured over Lhasa during the period of time when Trami was located over the Pacific Ocean. Tropical storm Jebi was able to pump air masses from the lower troposphere to near the tropopause along its track. Typhoon Utor played a key role in pumping up air masses when it made landfall.

Finally, we investigated the observed profiles with and without the influence of tropical cyclones. The relative frequency distribution of air parcels sampled over Lhasa originating from the lower troposphere show that maximum impact by typhoon is found at an altitude layer of $14.5-17$ km ($365-375$ K). Our findings confirm that air masses that originated from the Western Pacific region can also contribute to the composition of the ASM anticyclone as shown in several previous model studies (Park et al., 2009; Chen et al., 2012; Bergman et al., 2013; Vogel et al., 2014; Tissier and Legras, 2016). Our results also demonstrate that typhoons before landfall have a significant influence on ozone near the tropopause layer. Under these conditions, typhoons cause a low ozone value ($50-80$ ppbv) near the layer of $370-380$ K. However, typhoons during landfall have a strong impact on air masses in the middle troposphere. In our study, the transport pathways from air masses uplifted by tropical cyclones into the ASM anticyclone were identified. Moreover, their impact on the ozone values sampled over Lhasa in the ASM anticyclone during August 2013 was quantified.

In the TP region, there is a lack of dense in-situ meteorological observations, which might increase the uncertainties of meteorological reanalysis data, especial for the vertical velocity (Randel and Jensen, 2013). These uncertainties in vertical velocities impact the results of the CLaMS trajectory calculation. In addition, due to the lack of a detailed convective scheme in CLaMS, especially the vertical transport in small-scale convective clouds is underestimated. However, there is a lower limit for longer timescale transport over several days or weeks. For example, transport from the marine boundary layer to near the tropopause layer in Lhasa takes more than one week according to the results of the CLaMS trajectory calculation. Every year, about one third of tropical cyclones form in the tropical and subtropical Western Pacific (Matsuura et al., 2003). Due to the limited number of ozone profiles from Lhasa, it is still unclear how important the upward transport by tropical cyclones is. Therefore, it is necessary to quantify the contribution of the Western Pacific marine boundary layer air with uplifting and long-range transport in controlling the chemical composition of the upper troposphere within the ASM anticyclone. Case studies are certainly a first step toward understand the interplay between the ASM anticyclone and tropical cyclones, but further research is required to explore the quantification and intensities using long-term data records of ozone and water vapour. Only a small





amount of in-situ measurements in the region of the ASM anticyclone are so far available. Therefore, future balloon and aircraft measurements campaigns should focus on this region.

*Acknowledgements.* Ozone and water vapour data are from the SWOP campaign, which is funded by the National Natural Science Foundation of China (91337214 and 41675040). CALIPSO data used in this study can be found at https://eosweb.larc.nasa.gov/project/calipso/
5   calipso_table. This work was supported by the Chinese Scholarship Council and the German Academic Exchange Service providing the Sino-German (CSC-DAAD) Postdoc Scholarship Program.



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



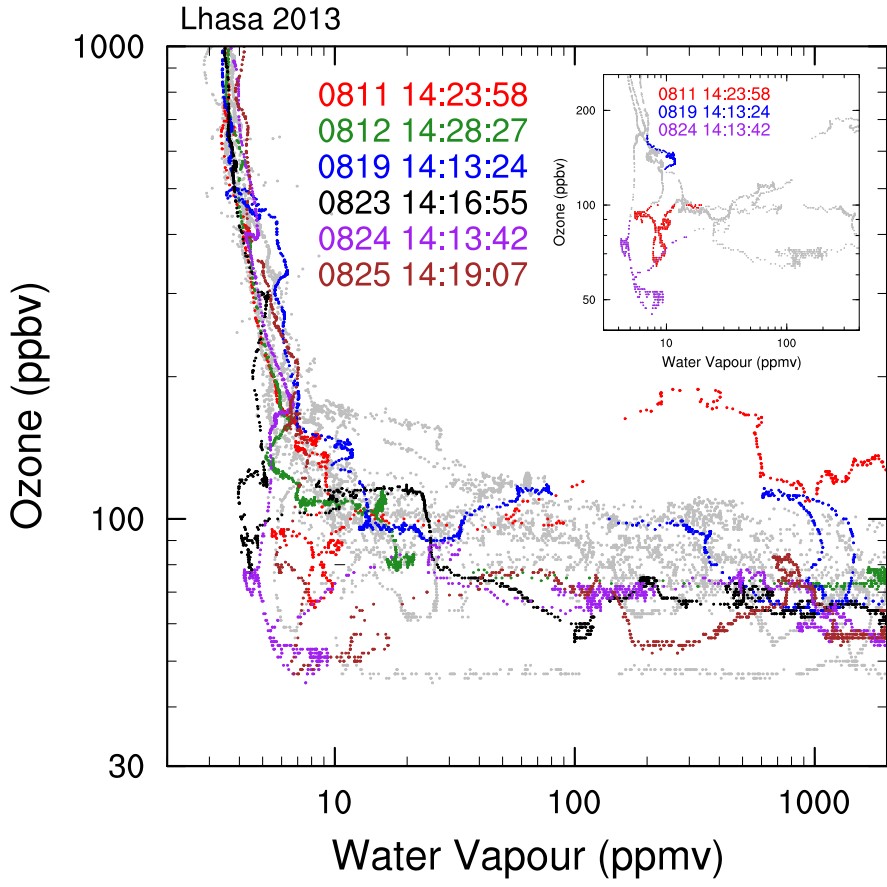

**Figure 1.** $H_2O-O_3$ correlations for balloon profiles measured over Lhasa in August 2013 (grey colour). The profiles measured on $11-12$ August were impacted by tropical storm Jebi and measurements on 19 and $23-25$ August were impacted by typhoon Utor or Trami. The layers of three profiles influenced by tropical cyclones are highlighted (inset).



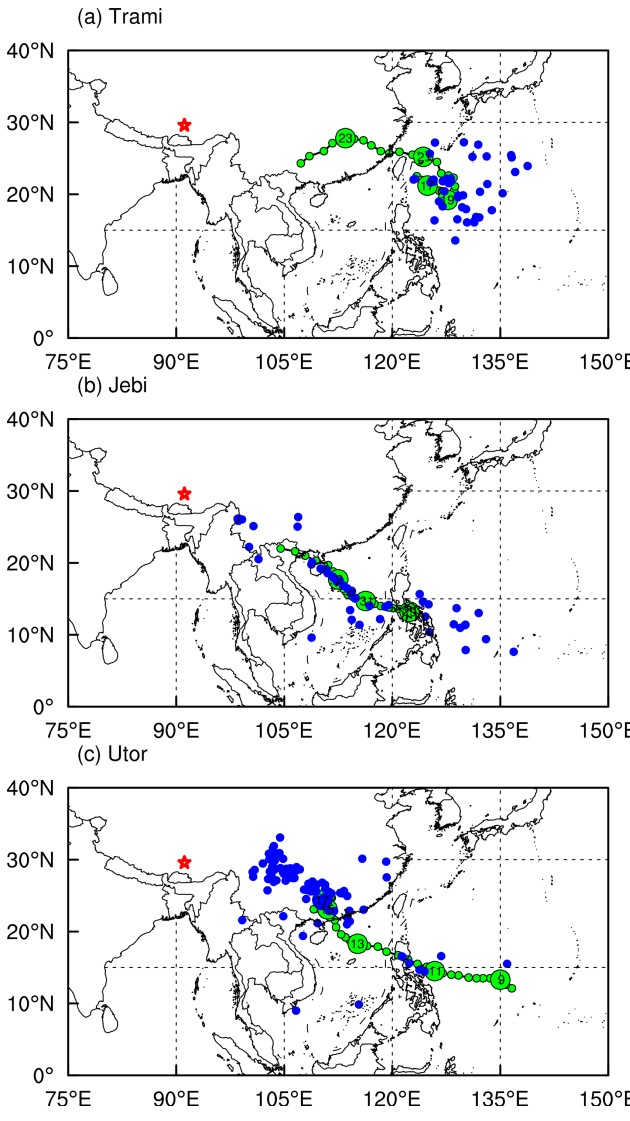

**Figure 2.** Pathways of tropical cyclones of (a) Trami, (b) Jebi, and (c) Utor are marked as green dots for every 6 hours. The number printed inside the large green dots indicates the day in August 2013, except 29 and 31 in Fig. 2b which refer to 29 and 31 July. The blue points denote the average geographical position where air parcels experienced strong uplift within tropical cyclones. The red star marks the location of Lhasa.



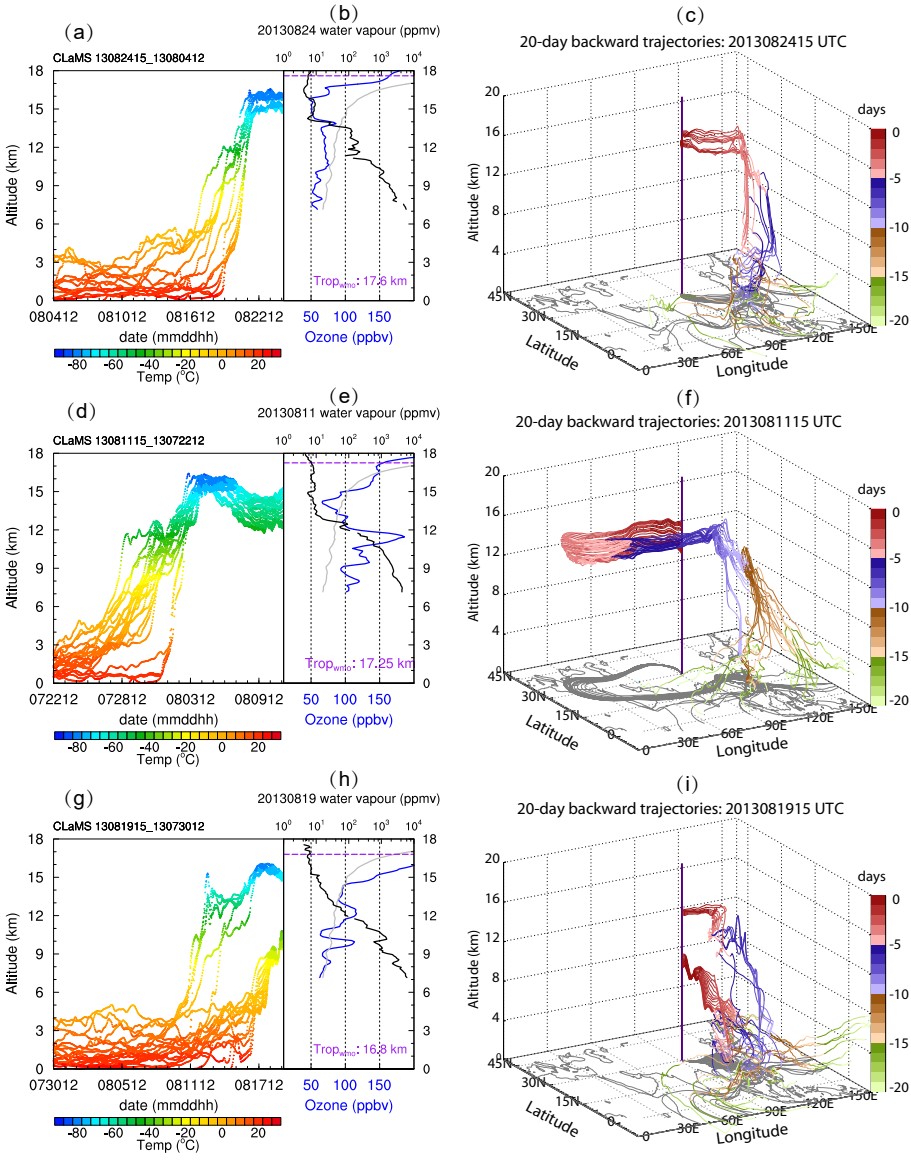

**Figure 3.** 20-day backward trajectories started along measured balloon profiles of target parcels on 24 (a and c), 11 (d and f), and 19 (g and i) August 2013. Backward trajectories influenced by tropical cyclones Trami, Jebi, and Utor are shown colour-coded by temperature (left). The vertical profiles of ozone (blue line), water vapour (black line), and the mean of ozone profiles in August 2013 (grey line) are also shown (middle). The geographical position (latitude, longitude, and altitude) of the 20-day backward trajectories is given (right) colour-coded by days observed from measurement. The vertical line marks the location of the Lhasa site. The grey lines in the maps show the longitude−latitude projection of the trajectories.



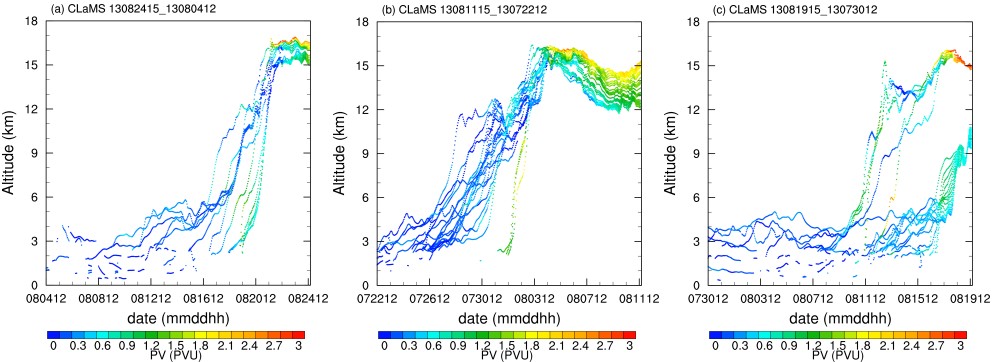

**Figure 4.** The same format as Fig. 3 (left), but potential vorticity (1 PVU=$10^6$ K m$^2$ kg$^{-1}$ s$^{-1}$) is shown along 20-day backward trajectories for observations impacted by (a) Trami, (b) Jebi, and (c) Utor.

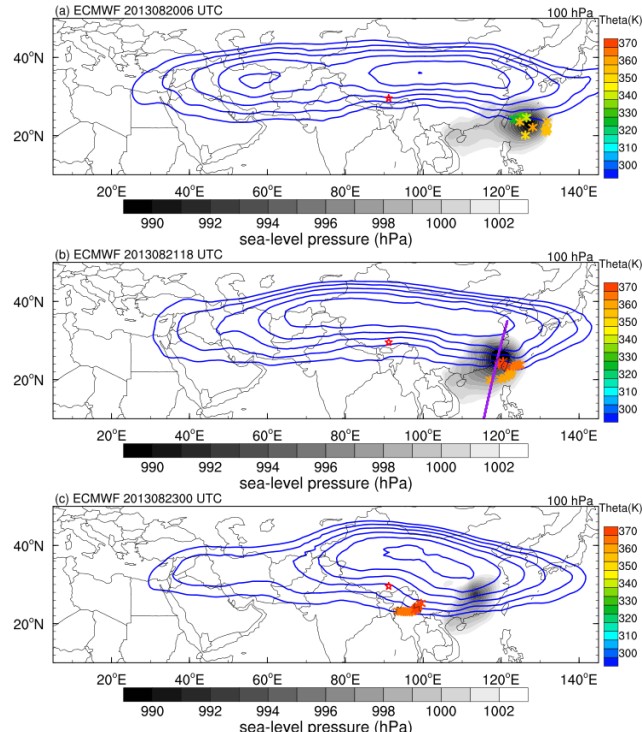

**Figure 5.** The geopotential height at 100 hPa pressure level (blue contour lines, $> 16.72$ km) and sea-level air pressure (shade, hPa) of Trami at (a) 06:00 UTC 20 August, (b) 18:00 UTC 21 August, and (c) 00:00 UTC 23 August. The asterisks, colour-coded by potential temperature, mark the geographical positions of target parcels. The purple line (middle panel) marks the CALIPSO orbit track. The red star marks the position of Lhasa.





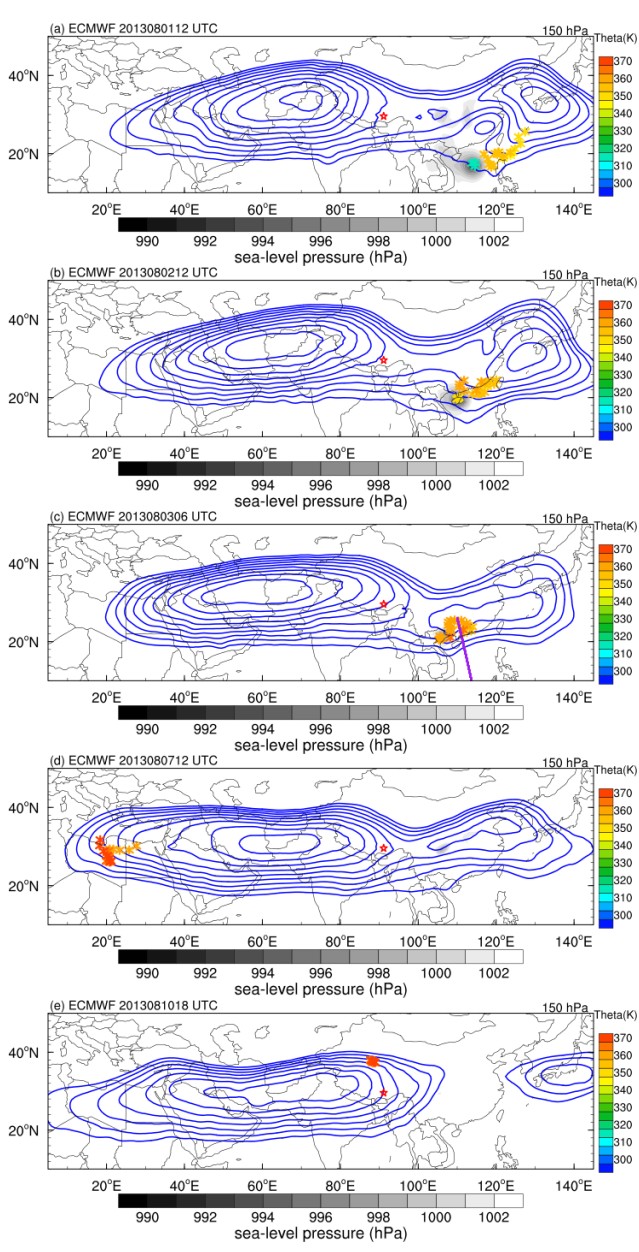

**Figure 6.** The same format as Fig. 5, but for Jebi. Here the geopotential height is shown at 150 hPa pressure level





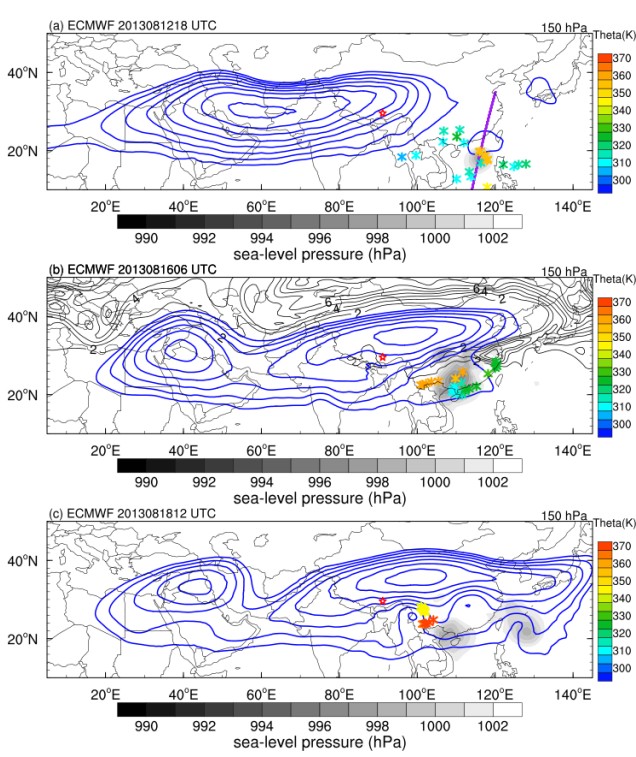

**Figure 7.** The same format as Fig. 6, but for typhoon Utor. In addition, PV isolines ($> 2$ PVU, solid black lines) are shown in Fig. 7b





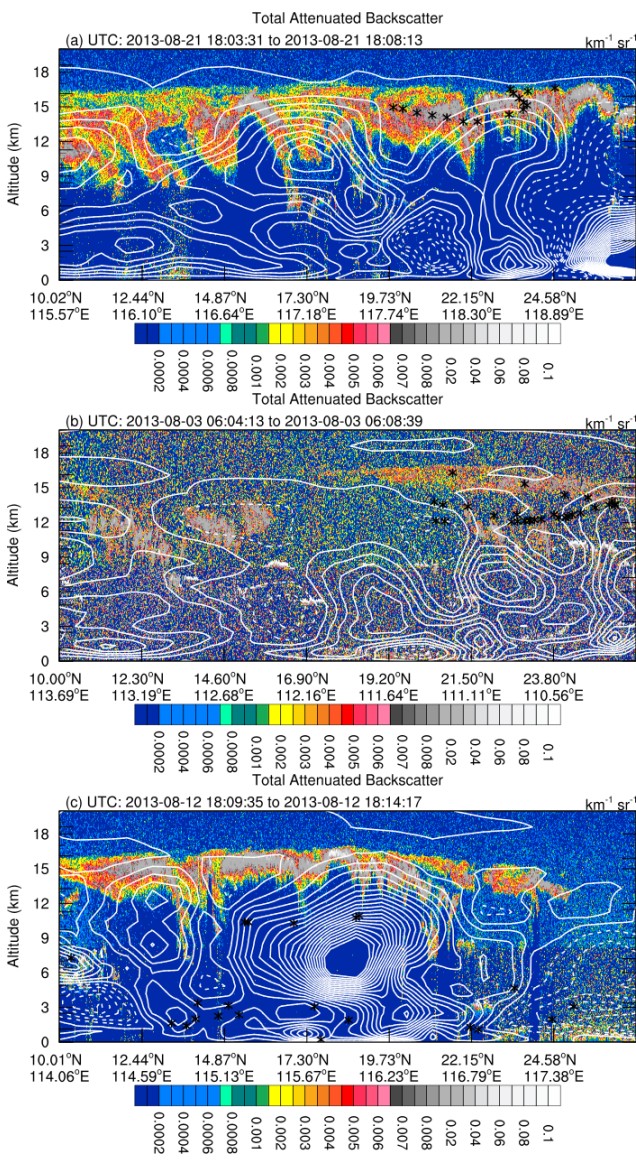

**Figure 8.** Total attenuated backscatter along CALIPSO orbit tracks of Figures 5b, 6c, and 7a. The asterisks mark the near locations of parcels with corresponding time.





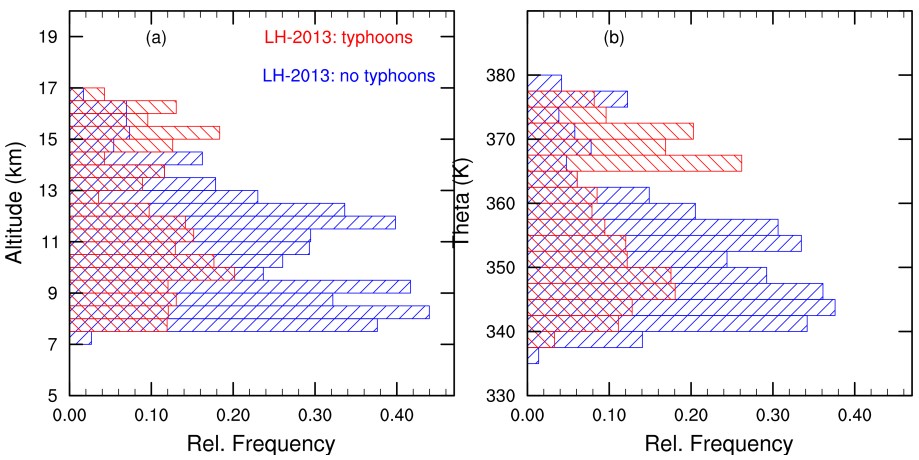

**Figure 9.** The relative frequency distribution of the target air masses originating from the boundary layer with the influence of typhoons (red) and without the influence of typhoons (blue) versus (a) altitude and (b) potential temperature in August 2013.





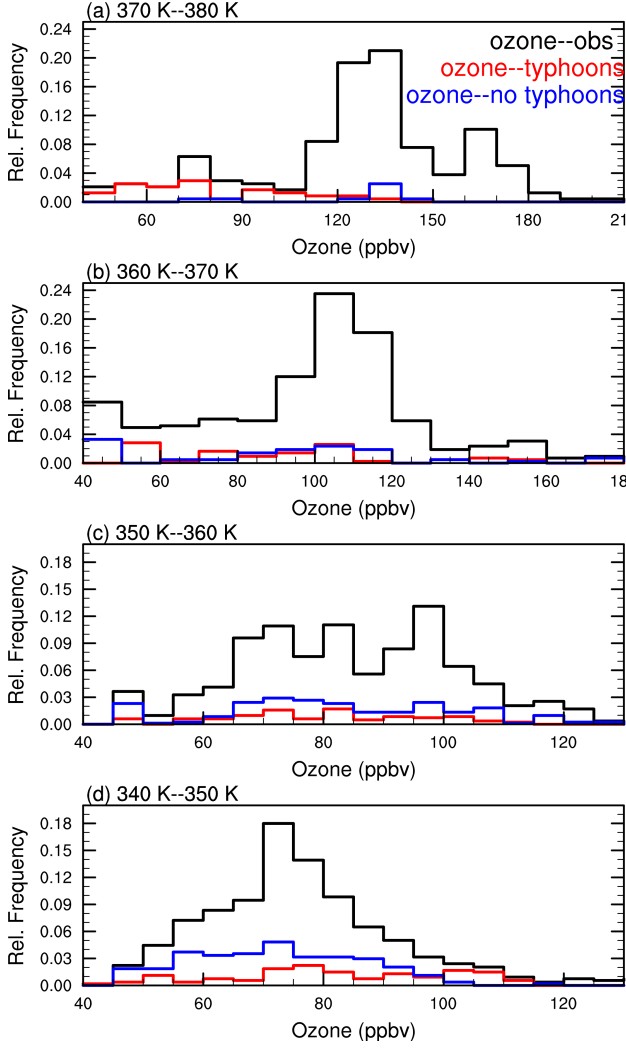

**Figure 10.** The relative frequency distribution of all ozone profiles observed in 2013 (black) and ozone with (red) or without (blue) the influence of tropical cyclones with respect to potential temperature layer of (a) 370−380 K, (b) 360−370 K, (c) 350−360 K, and (d) 340−350 K.





**Table 1.** Flight statics for balloon launches during the August 2013 SWOP campaign in Lhasa

| No. | Launch time (UTC) | | Burst altitude (km) | Influenced by tropical cyclones |
|---|---|---|---|---|
| lh022 | 20130804 | 14:09 | 36.59 | |
| lh023 | 20130805 | 14:54 | 36.69 | |
| lh024 | 20130806 | 14:06 | 21 | |
| lh025 | 20130807 | 14:34 | 36.27 | |
| lh026 | 20130808 | 14:41 | 36.36 | |
| lh027 | 20130809 | 15:10 | 31.39 | |
| lh028 | 20130810 | 14:20 | 36.30 | |
| **lh029** | **20130811** | **14:23** | **26.56** | **Jebi** |
| lh030 | 20130812 | 14:28 | 33.62 | Jebi |
| lh031 | 20130813 | 14:29 | 35.14 | Jebi |
| lh032 | 20130814 | 14:18 | 36.32 | |
| lh033 | 20130815 | 14:41 | 33.28 | |
| lh034 | 20130816 | 14:22 | 35.77 | |
| lh035 | 20130817 | 14:14 | 31.52 | Utor |
| lh036 | 20130818 | 14:12 | 34.82 | Utor |
| **lh037** | **20130819** | **14:13** | **34.81** | **Utor** |
| lh038 | 20130820 | 14:17 | 29.66 | Utor |
| lh039 | 20130821 | 14:12 | 32.48 | Utor |
| lh040 | 20130822 | 14:18 | 35.39 | Utor |
| lh041 | 20130823 | 14:16 | 32.47 | Utor, Trami |
| **lh042** | **20130824** | **14:13** | **34.07** | **Utor, Trami** |
| lh043 | 20130825 | 14:19 | 32.20 | Utor, Trami |
| lh044 | 20130826 | 14:37 | 18.39 | Utor, Trami |
| lh045 | 20130827 | 14:16 | 18.82 | |

Flights in bold are shown in detail.

**Table 2.** Classification of tropical cyclones: Jebi, Utor, and Trami in 2013

| Name | Duration | Intensity | Peak | SSHWS |
|---|---|---|---|---|
| Jebi | Jul.28−Aug.3 | 985 hPa | 95 km/h | Tropical storm |
| Utor | Aug.8−18 | 925 hPa | 195 km/h | 4 |
| Trami | Aug.16−24 | 965 hPa | 110 km/h | 1 |