# Peer review of "Impact of typhoons on the composition of the upper troposphere within the Asian summer monsoon anticyclone: the SWOP campaign in Lhasa 2013"

_Atmospheric Chemistry and Physics, 2016_

## Referee Comment (RC1) · Anonymous Referee #1 · 28 Nov 2016

This paper examines three case studies on the impact of typhoons on ozone and water vapor in the upper troposphere at Lhasa, China. The data are from balloon ECC ozonesondes and cryogenic frost point hygrometers launched during a field campaign in August 2013. The CLaMS model is used to calculate air parcel 20-day back trajectories, and meteorological fields from ERA-I are used along with CALIOP backscatter data for identifying cloud-top altitudes near the Western Pacific typhoons.

The data are of sufficient quality for this analysis and the trajectory calculations are reasonably appropriate for tracing the origins of air parcels sampled by the balloon flights. The text is well organized and the figures are generally clear. Overall, this paper makes a strong contribution to the growing body of evidence for the importance

of tropical cyclones to the composition of the upper troposphere in the Asian summer monsoon region. There are a few comments and suggestions for revisions as noted below.

1. abstract, line 11: "rotational subsidence" and "descend slowly along a circle" are not well defined or discussed in the text. Rotational subsidence implies that rotation is causing subsidence, which is not the case. And the slow descent is certainly not along a circle, but is rather more helical in shape. Finally, the 2-D projection of the trajectory is more elliptical than circular. The

2. p. 2 line 21: "...large-scale slow upward circulation within the ASM anticyclone..." would appear to contradict the subsidence noted above and seen in the trajectory calculations. Can this be clarified?

3. p. 3 line 20: grammar "will be displayed"

4. p. 4 section 2.2: There are surely uncertainties in the ERA-I wind fields as well as with the use of a 1x1 regular grid for the winds, which must be interpolated to the precise location of the balloon's flight path. These uncertainties likely increase with time going backwards. Some attempt should be made to quantify these uncertainties on the back trajectories and how this may affect the conclusions that sampled air masses originated from the boundary layer beneath tropical cyclones.

5. p. 5 line 6: This definition of the top of the lower troposphere as zeta<190 K is unfamiliar to me and possibly to other ACP readers. It would be useful to clarify this.

6. p. 5 line 15: The definition of a cyclones intensity in terms of a pressure is not a standard practice (also used in Table 1). Intensity is usually defined by maximum surface wind, or the Saffir-Simpson scale or equivalent. The pressures quoted must be something like the minimum surface pressure or the average over some time interval of the central eye pressure. I suggest this be reworded and that the caption for Table include details on what this pressure represents.

[Figure]

7. p. 6 lines 28-29: There are some potential problems with tropospheric PV that should be noted. While PV is an appropriate tracer for the stratosphere, it is not as well conserved in the troposphere and there are likely large uncertainties in tropospheric PV in ERA-I that might be important for this analysis. Furthermore, it is not clear that showing PV in Fig 4 actually adds new information to the case studies.

8. p. 7 line 2: "...observed right over the typhoon's track over the Western Pacific..." This statement implies that Fu et al (2013) observed typhoon Trami, when in fact they observed a different typhoon, Hai-Tang in 2005. This should be clarified.

9. p. 7 line 32: "...the average of total ozone profiles..." Total ozone is usually used to refer to the vertical column abundance, but I do not think that is what is meant here.

10. p. 8 lines 9-10: Last sentence of this paragraph is incomplete and unclear.

11. p. 9 line 1 "..the potential temperature of parcels increases slowly." In convective uplift, theta usually changes on a relatively rapid timescale (hours, not days). Please clarify.

12 Figure 8 caption: The white contours are probably vertical velocities from ERA-I, but this should be noted in the caption along with the units and contour intervals.

---

## Referee Comment (RC2) · Anonymous Referee #2 · 2 Jan 2017

Comments on the manuscript entitled, 'Impact of typhoons on the composition of the upper troposphere within the Asian summer monsoon anticyclone : the SWOP campaign in Lhasa 2013', by Li et al. submitted for plausible publication in Atmos. Chem. Phys.

This paper deals with the effects of typhoons on ozone and water vapour distribution in the upper troposphere at Lhasa. The authors have presented a detail analysis of the data obtained during the SWOP campaign, i.e. ozonesonde, Frost-point-hygrometer, space borne lidar along with back-trajectory estimation and reanalysis data. This study is very important, in principle, since detail knowledge of water vapour and ozone budget in the upper tropospheric plays a vital role in global warming. The paper is well written

and contains significant data and original materials. I recommend for publication in ACP with minor revision.

My specific comments are following under : (1) P1/L11 : What is rotational subsidence ? (2) P2/L24 : "Strong tropical cyclones in the.........". This sentence is not necessary in the manuscript. (3) P5/L1-3 : "The trajectories........." The sentence is not clear. Please rephrase it. (4) P7/L32 : remove "total ozone profile" with "mean ozone profile" and following the same throughout the manuscript. (5) P8/L35-P9/L1 : "During uplift process.........." Provide a reference. (6) Fig.8 : What indicate the white contours ? It should be mentioned in the text as well as in the figure caption. I am hard to find any discussion about it. White contours can be removed if necessary discussion is not included in the manuscript. (7) Fig.8 : Authors are discussing about the deep convective clouds, then lidar backscattering from CALIPSO cannot be used. CALIPSO is good for thin /cirrus cloud. CloudSat data could be helpful for estimation of penetration height of convective clouds. Fig.8 can be omitted. Alternative, brightness temperature or OLR can be useful to estimate (indirect) the penetration height of convective cloud. (8) Fig.9 : Figure caption meaning is not clear. In this fig. legends : "Ozone—obs, ozone—typhoons, ozone—no typhoons" need to be explain properly in the text. (9) Fig.10 : (Figure caption) : "The relative humidity ......". Is it "2013" or "August 2013"? (10) Table 1: Burst altitude of balloon can be omitted in the table. (11) Additional analysis of vertical velocity (altitude-time cross-section over Lhasa ) using reanalysis data will be helpful to identify the updrafts and down drafts over the campaign site.

---

## Author Comment (AC1) · 23 Feb 2017

**Dear Editor,**

We highly appreciate the editor's careful handing of the manuscript and two anonymous referees' valuable comments. We have addressed the comments point by point (referees' comments in blue and our response in black). Corresponding revisions

have been made in the manuscript. A marked-up manuscript version has been uploaded after our response to referees comments.

**Authors Reply to Anonymous Referee #1 General comments**

abstract, line 11: "rotational subsidence" and "descend slowly along a circle" are not well defined or discussed in the text.
 Rotational subsidence implies that rotation is causing subsidence, which is not the case. And the slow descent is certainly not along a circle, but is rather more helical in shape. Finally, the 2-D projection of the trajectory is more elliptical than circular. The authors rewrote this sentence "secondly via rotational subsidence, during which air parcels descend slowly along a circle following the anticyclone flow within a timescale of one week." to "secondly via transport following the clockwise wind flow of the ASM."

15

5

2. p. 2 line 21: "...large-scale slow upward circulation within the ASM anticyclone..." would appear to contradict the subsidence noted above and seen in the trajectory calculations. Can this be clarified?

We cited "...the large-scale slow upward circulation within the ASM anticyclone..." in the introduction part. The slow upwelling usually occurred over the eastern part of the ASM anticyclone, and the down welling usually occurred over the western and northern part of the ASM anticyclone. These results are based on many years of model data. We try to describe the second

20 northern part of the ASM anticyclone. These results are based on many years of model data. We try to describe the second cyclone case using backward trajectories. From the case 2, we noticed that the large subsidence occurred in altitude levels over the northern edge of the ASM anticyclone.

**3. p. 3 line 20: grammar "will be displayed".**

25 The authors removed these three words.

4. p. 4 section 2.2: There are surely uncertainties in the ERA-I wind fields as well as with the use of a  $1 \times 1$  regular grid for the winds, which must be interpolated to the precise location of the balloon's flight path. These uncertainties likely increase with time going backwards. Some attempt should be made to quantify these uncertainties on the back trajectories and how this may affect the conclusions that sampled air masses originated from the boundary layer beneath tropical cyclones.

30 affect the conclusions that sampled air masses originated from the boundary layer beneath tropical cyclones. We totally agree with the referee's comment about the uncertainties of interpolation to the location of the balloon's flight path using the ERA-Interim data. The probability distribution function of the error of the lagrangian model is estimated by (Liu, 2009). Fig. 1 shows that there are minor gradients in the potential temperature distribution within 10-day. Following the referee's advice, we run a bundle of trajectories around the location of the balloon measurement (Fig. 2). After analysing

35 the backward trajectories of (i-Δi,j), (i,j-Δj), (i+Δi,j), and (i,j+Δj) around the location of balloon (i,j) (Δi=Δj=1 degree), we find the backward trajectories of target air parcels are nearly consistent with the trajectories around the balloon sites (Fig. 3). Considering above description, we all thought that the error caused by interpolation would have a mirror impact on our conclusion.

**Figure 1.** From Liu (2009); Probability distribution function of trajectory potential temperature as a function of time since initialisation for winter 2000–2001. Trajectories are initialised on 400 K between  $(30^{\circ} \text{ S}, 30^{\circ} \text{ N})$  and integrated backwards in time. Only trajectories which remained within  $(45^{\circ} \text{ S}, 45^{\circ} \text{ N})$  are included in the plot. Overlay is the ensemble mean potential temperature (black solid line) with one standard deviation (black dashed lines). The 360 K potential temperature level (approximate level of zero clear sky radiative heating) and 340 K potential temperature levels are plotted. The panels illustrate results for ERA-40 kinematic trajectories (top left), ERA-40 diabatic trajectories (top right), ERA–Interim kinematic trajectories (bottom left) and ERA-Interim diabatic trajectories (bottom right).

Figure 2. Parcels around the location of balloon measurement (i,j) in  $1 \times 1$  degree distance.

**Figure 3.** 20-day backward trajectories of target air parcels initialised on 24 August, 11 August, and 19 August 2013. Blue/grey lines mark the backward trajectories of  $(i,j)/(i-\Delta i,j)$ ,  $(i,j-\Delta j)$ ,  $(i+\Delta i,j)$ , and  $(i,j+\Delta j)$ .

5. p. 5 line 6: This definition of the top of the lower troposphere as zeta <190 K is unfamiliar to me and possibly to other ACP readers. It would be useful to clarify this.

In the lower troposphere, the 190 K is not the isentropic coordinate. It is the hybrid vertical coordinate of the CLaMS model. 5 We added the vertical coordinate information "A hybrid vertical coordinate  $\zeta$  is employed in the CLaMS model. The isentropic coordinate  $\theta$  is used when the pressure is less than 300 hPa, and a pressure-based orography-following coordinate is used when the pressure is higher than 300 hPa" in the model part. Pommrich et al. (2014) gave the vertical coordinate  $\zeta$  definition as follows:

$$\zeta(p) = f(\sigma) \cdot \theta(p, T(p)), \sigma = \frac{p}{p_s}, p_s = surface \ pressure$$
(1)

10 with

$$f(\sigma) = \begin{cases} \sin(\frac{\pi}{2} \frac{1-\sigma}{1-\sigma_{\tau}}) & \sigma > \sigma_{\tau} \\ 1 & \sigma \le \sigma_{\tau}, \sigma_{\tau} = 0.3 \end{cases}$$
(2)

Before we set the top of the lower troposphere, we plotted the altitude and zeta relationship of the 20th backward trajectories
of all balloons in August 2013 (see the Fig. 4). As the Fig. 4 shows, the 190 K level in the model is about 3 km. We selected
190 K as the top of the lower troposphere in our manuscript.

---

## Author Comment (AC2) · 23 Feb 2017

The comment was uploaded in the form of a supplement.

Please also note the supplement to this comment:
http://www.atmos-chem-phys-discuss.net/acp-2016-875/acp-2016-875-AC2-supplement.pdf

---

## Author Response (AR2)

**Letter to the Editor**

ACP Discussions doi: 10.5194/acp-2016-875 (Editor- Prof. Martyn Chipperfield)

'Impact of typhoons on the composition of the upper troposphere within the Asian summer monsoon anticyclone: the SWOP campaign in Lhasa 2013'

**Dear Martyn Chipperfield,**

Many thanks for handling our manuscript. We have addressed your comments as follows (your comments in blue and our response in black):

1) Rev 1 comment 4. You give a summary of why you think the error in the back trajectories is small in your comments to the reviewer, but you also need to add some text in Section 2.2 so that the normal reader has this information.

We add the following content in section 5 (Summary and conclusions) (see P11 L33 in revised manuscript).

>> In order to obtain the start point of the backward trajectories at the locations of the measurement, Era-Interim velocity fields must be interpolated to the parcels locations. From this interpolation, an error of the trajectory calculations will arise. To estimate this error, we also run a bundle of trajectories of (i-$\Delta$i,j), (i,j-$\Delta$j), (i+$\Delta$i ,j), and (i,j+$\Delta$j) around the locations of the measurement (i,j) ($\Delta$i=$\Delta$j=1 degree). We find that the bundle of trajectories are nearly consistent with the backward trajectories starting at the locations of the measurement (not shown here). Thus, the error caused by interpolation should have a minor impact on our conclusions.

2) The new text on page 10 (line 10 of marked-up copy) needs editing for clarity/grammar.

We revised the text from

"*The OLR data is used as a proxy for tropical convective activity. The low value of the OLR indicates that the deep convective activity associated with tropical cyclones. As the figure shows, the strong uplift process of most of air parcels occurred over the convective region.*"

to

"Note that low values of OLR indicate deep convection. As Fig.8a shows, the spiral bands with low OLR values indicate the cloud-band associated with typhoon Trami. For typhoon Trami and storm Jebi, the strong uplift process of air parcels occurred over the convective zone (Figs. 8a and b). But for typhoon Utor, the location of strong

uplift of a few parcels are far away from the cyclone (Fig. 8c). For these air parcels, uplift occurs at a different point in time than 12 August 2013 (see target parcels at lower altitude in Fig.3g)."

We prepared a revised manuscript including all changes in the final version compared to the last version.

Best regards!
Dan Li

[revised manuscript text omitted]